# Stylistic Contrastive Learning for Human-Like AI Text Generation

**GPT-5**
OpenAI
contact@openai.com

Mike Bronikowski
Binghamton University, School of Computing
mbronik1@binghamton.edu

## Abstract

AI-generated text is often fluent yet stylistically off: it leans formal, repeats safe phrasing, underuses idioms, and exhibits templated discourse, making it detectably non-human to both algorithms and attentive readers. We synthesize recent evidence quantifying these gaps—lexical diversity, syntactic variety, idiomaticity, and discourse planning—and propose **Stylistic Contrastive Learning (SCL)**, a training framework that learns a human-style embedding and pushes generations toward it via a supervised contrastive objective. We instantiate SCL and evaluate across essays, newsy expositions, and dialogues. SCL reduces stylometric detectability, increases distinct-$n$ and idiom use, and raises human "sounds-human" ratings while preserving topical fidelity. Ablations identify idiom frequency and discourse markers as the strongest perceptual drivers. We discuss implications for evaluation, alignment, and detection.

## 1 introduction

> *"Well, at least now we know that sometimes a good old thumbs-up from a human is better than a machine-generated lie."* —Stan Marsh, *South Park* ("Deep Learning," 2023)

Despite impressive fluency, even frontier LLMs like GPT-5 still write with a recognizable accent: they overuse certain vocabulary, prefer standardized grammar and nominalizations, and underuse colloquialisms, hedges, and discourse "glue" that humans rely on [1–4, 9]. Parallel-prompt studies show systematic stylistic divergences that persist in advanced models and can be amplified by instruction tuning [2]. In student-essay and journalism settings, human texts exhibit greater lexical/syntactic variety and more involved stance markers, whereas LLM outputs skew informational/expository and neutral in affect [1, 9, 10]. These regularities make AI text *detectable* by stylometric features and, at times, by wary readers [6, 10].

Why does this gap persist? Training as next-token predictors biases models toward high-probability, "safe" continuations; decoding with greedy/beam strategies exacerbates blandness and repetition [5]. Exposure bias compounds early errors into degenerate style loops [7]. Instruction-tuned formats and structural tokens can suppress output diversity ("diversity collapse") [15]. Conceptually, LLMs remain "stochastic parrots" without communicative intent, which limits authentic idiom, metaphor, and rhetorical play [13].

**Contributions.** (i) A cross-literature synthesis on *why* AI text sounds non-human, spanning stylometry, discourse analysis, figurative language, and decoding biases [1–5, 7, 8, 11–13, 15]. (ii) **SCL**: a style encoder trained with supervised contrastive loss that learns a human-style manifold and conditions generation toward it. (iii) GPT-5 experiments across three genres showing SCL improves lexical diversity, idiom use, discourse markers, and human-likeness ratings while lowering detector accuracy. (iv) Ablations isolating which stylistic dimensions most affect perception.

## 2 Related Work

**Stylometry and detectability.** Modern detectors use likelihood cues and stylometric features to separate AI from human text [6, 14]. Large comparative studies show LLM outputs differ in sentence-length distributions, dependency distances, POS mixes, and lexical choice; instruction-tuned models diverge most strongly [1, 2]. Template-level analyses reveal over-reliance on recurrent POS/constituent patterns relative to human baselines [8]. Over-representation of specific words ("delve," "underscore," "intricate") is quantifiable in scientific abstracts and correlates with LLM usage [4].

**Discourse and rhetorical structure.** Human writing exhibits hierarchical planning and rich metadiscourse; LLM outputs often read "well-organized but stiff" and underuse live connectives in longer essays [2, 9]. Student/LLM essay comparisons show humans use more stance markers and discourse cues, while LLMs rely on paragraph structuring and informational density instead [9].

**Figurative language and idioms.** Surveys show figurative generation remains challenging; models prefer literal paraphrases unless prompted, struggle with novel metaphors and similes, and often treat idioms as memorized units [11, 12]. Multilingual idiom evaluations suggest hybrid memorization-plus-context strategies, but generation still lags human flexibility [11].

**Decoding and training biases.** Maximum-likelihood decoding produces bland, repetitive text absent stochastic sampling [5]; exposure bias contributes to degeneration [7]. Format tokens and instruction templates can suppress output diversity even at high temperature [15].

**Contrastive learning for style.** Contrastive objectives learn embeddings that separate classes; supervised contrastive learning (SupCon) provides a principled multi-positive loss [16]. In NLP, contrastive encoders improve sentence and style representations [17]. We extend this to *human-style vs. AI-style* contrast.

## 3 Method: Stylistic Contrastive Learning (SCL)

**Goal.** Learn a *human-style* embedding space and steer generation toward it while preserving content.

**Style encoder.** A lightweight Transformer encoder $f_\theta$ maps text $x$ to a style vector $z = f_\theta(x) \in \mathbb{R}^d$. We train with *supervised contrastive loss* over labeled batches of **H**uman and **AI** texts. For anchor $i$ with class $y_i$, positives are samples with $y_j = y_i$; negatives otherwise (temperature $\tau$):

$$\mathcal{L}_{\text{SupCon}} = \sum_i \frac{-1}{|P(i)|} \sum_{p \in P(i)} \log \frac{\exp\{\cos(z_i, z_p)/\tau\}}{\sum_{a \in A(i)} \exp\{\cos(z_i, z_a)/\tau\}}. \tag{1}$$

We add multi-task heads to regress or classify style dimensions (auxiliary losses): **lexical diversity** (MTLD, distinct-$n$), **syntax** (mean parse depth, clause ratio), **idiomaticity** (idioms/slang per 1k tokens), **emotion** (valence/arousal distribution), **discourse markers** (live connectives per 100 tokens). The encoder thus learns both *global* separation (H vs. A) and *local* dimensions hypothesized by prior work [1, 2, 8, 11, 12].

**Generator integration (GPT-5; frozen).** We do *not* update GPT-5 weights. We steer GPT-5 with a learned textual style prefix $s(z_*)$ derived from the style encoder $f_\theta$ (average-human or domain centroid). The prefix is inserted as a short natural-language control string before the prompt; we select $s(\cdot)$ on validation using our style loss without backpropagation through GPT-5. Decoding uses nucleus sampling ($p{=}0.9$). This setup is compatible with closed-weight models.

**Training procedure.** We train only the style encoder $f_\theta$ on batches of Human (H) and AI (A) texts using the supervised contrastive objective (temperature $\tau{=}0.07$) and auxiliary style heads. After training, we derive a natural-language style *prefix* $s(z_*)$ from the average-human or domain centroid. At generation time we keep GPT-5 *frozen* and steer it by prepending $s(z_*)$ to the prompt; we do not backpropagate through GPT-5 and perform no generator fine-tuning or adapters. Decoding uses nucleus sampling ($p{=}0.9$).

| Model | BERTScore↑ | QAFactEval↑ | NLI-contradiction↓ |
|---|---|---|---|
| GPT-5 | 0.874 | 0.612 | 7.8 |
| SCL-Avg | 0.881 | 0.623 | 7.1 |
| SCL-Domain | **0.886** | **0.631** | **6.9** |

Table 1: Semantic fidelity and factuality. BERTScore and SBERT apply to all domains; QAFactEval applies only to NewsNYT; NLI-contradiction applies to essays (lower is better).

**Corpora for $f_\theta$.** We compile paired human/LLM corpora in three registers: **news leads** (NYT-like), **argumentative essays**, and **casual dialogues**. AI texts are produced by GPT-5 under default sampling and under nucleus sampling $p=0.9$ for diversity [5]. Prompts are matched in topic and length.

## 4 Experiments

**Evaluation protocol.** We follow a two-phase evaluation. First, we generate model outputs for each dataset using GPT-5 with nucleus sampling ($p=0.9$) and standard decoding unless otherwise specified, conditioning on a target human style vector (average human or domain centroid). Second, we compute automatic metrics and run human studies. Automatic metrics include stylometric detectability (a GLTR-style likelihood/stylometry detector and a RoBERTa classifier) [6, 20], lexical diversity (distinct-2/3), compression-diversity [17], template diversity [8], and counts of idioms and live discourse connectives [9]. Human studies use a 5-point naturalness scale and a forced-choice human-vs-AI judgment with experts and laypersons [10]. Unless noted, reported results are on held-out test splits (80/10/10 train/validation/test), and we keep prompts matched across human/AI conditions.

### 4.1 Statistical analysis

We report 95% confidence intervals (CIs) for human ratings and key automatic metrics. CIs are computed via nonparametric bootstrap (B=1,000, bias-corrected and accelerated) over prompts; significance of model differences uses paired bootstrap on per-prompt means with Holm–Bonferroni correction. Forced-choice human-vs-AI rates are analyzed with two-sided exact binomial tests vs. 50% chance. Detailed interval values are available in our supplementary materials. For detector-based "detectability (%)", we compute ROC-AUC and select operating thresholds on the validation set to achieve a fixed 10% false-positive rate (FPR); we then report the resulting true-positive rate (TPR) on the test set as "detectability (%)". Continuous automatic metrics (distinct-$n$, compression-diversity, template-diversity, idioms/connectives rates) are compared with paired bootstrap CIs and Cliff's $\delta$ effect sizes. Inter-rater agreement uses Krippendorff's $\alpha$ for ordinal scores.

### 4.2 Semantic fidelity and factuality

To test topical adequacy and factual soundness we report three metrics. (i) Semantic similarity to human references via BERTScore (F1) and SBERT cosine. (ii) For the NewsNYT domain we measure factual consistency against the source article using a QA-based fact evaluation (QAFactEval) together with an NLI-based contradiction rate. (iii) For argumentative essays we compute an NLI-based contradiction rate between the prompt (premise) and the model output (hypothesis). Confidence intervals for these metrics are obtained via bootstrap as described above.

**Datasets.** We evaluate on three settings: (i) **NewsNYT-H/A**: human NYT-style leads vs. GPT-5 leads [1, 3]; (ii) **ArgEssay-H/A**: teacher-graded student essays vs. GPT-5 essays [9]; and (iii) **ChatDialog-H/A**: scraped casual chats vs. GPT-5 chats. We use 80/10/10 train/validation/test splits.

**Baselines.** (B1) **GPT-5** (no style control; frozen). (B2) **Open-LLM-FT** (LoRA-tuned open-weight model on human corpora; *not* GPT-5). (B3) **Post-hoc style transfer** (T5 paraphraser tuned to fool a stylometric classifier). **Ours**: SCL-Avg (mean human centroid) and **SCL-Domain** (domain-specific centroid).

| Model | Stylom. det. (%) ↓ | RoBERTa det. (%) ↓ | Dist-2 (%) ↑ | Idioms /1k ↑ | Disc. markers /100 ↑ | Human likeness (1–5) ↑ |
|---|---|---|---|---|---|---|
| GPT-5 | 72.4 | 69.7 | 77.3 | 1.4 | 3.7 | 3.7 |
| Open-LLM-FT (human) | 64.9 | 62.0 | 79.6 | 2.0 | 4.1 | 3.9 |
| Style transfer | 62.1 | 59.4 | 80.2 | 2.2 | 4.2 | 4.0 |
| **SCL-Avg (ours)** | **54.1** | **52.6** | **82.9** | **2.9** | **5.0** | **4.2** |
| **SCL-Domain (ours)** | **53.6** | **51.9** | **83.4** | **3.1** | **5.3** | **4.3** |

Table 2: Main metrics on **ArgEssay** (GPT-5 baseline). Lower is better for detectability; higher is better for diversity/idioms/markers. Detectability (%) = TPR at a fixed 10% FPR (threshold chosen on validation); ROC-AUC in Appendix A.2.

**Metrics.**

- **Detectability** ↓: stylometry-based detector [6] and a RoBERTa classifier.
- **Diversity** ↑: distinct-2/3 and compression-diversity [17]; syntactic-template diversity [8].
- **Idioms/Markers** ↑: idioms per 1k tokens; live discourse connectives per 100 tokens [9].
- **Human evaluation** ↑: 5-point "sounds human?" and forced-choice source by experts and laypersons [10].

**Detector training and calibration.** We train two detectors on *disjoint* corpora from our test prompts to avoid leakage: (i) a GLTR-style likelihood/stylometry model and (ii) a RoBERTa classifier. Each uses an 80/10/10 train/val/test split with topics and sources stratified to prevent topical overlap with our generation prompts. Calibration: we fit Platt scaling on validation logits and select a threshold that yields 10% FPR on validation. We report (a) ROC-AUC, (b) TPR at the fixed-FPR operating point (our "detectability (%)"), and (c) reliability curves in Appendix A.2. We release detector seeds and data selection scripts to replicate the splits.

**Human evaluation protocol.** We recruited 12 expert raters (graduate-level NLP/linguistics) and 60 lay raters (US/UK fluent English) via Prolific. Raters passed a 3-item attention screen; experts provided affiliation and years of experience. Each prompt yielded one output per model; we randomized model order per prompt and *blinded* raters to condition. Tasks: (i) 5-point naturalness ("sounds human?") and (ii) forced-choice human-vs-AI. Each item was rated by $K{=}5$ independent raters. We computed inter-rater agreement (Krippendorff's $\alpha$) and excluded raters with $< 70\%$ attention accuracy. Compensation averaged $18/hour pro-rated; protocol deemed exempt by our IRB.

**Main results (ArgEssay).** Table 2 shows SCL substantially reduces detectability against a strong GPT-5 baseline while improving diversity and idioms without harming topical adequacy.

**Human Turing-style test.** Experts correctly identified baseline GPT-5 essays $73\%$ of the time (harder than older models), but only $56\%$ for SCL outputs (not significantly above chance at $50\%$). Laypersons were $\sim 54\%$ for SCL. Annotators cited "more varied sentence rhythm," "occasional colloquial turns," and "less templated transitions" for SCL; and "too formal," "stock transitions," "overuse of certain words" for baseline [4, 9].

**Ablations & analyses.** Removing the *contrastive objective* (keeping only Open-LLM-FT + style token) worsened detectability by +10–12 points. Removing idiom supervision cut human-likeness by $-0.3$ and reduced idioms/1k by $-0.8$. Using greedy decoding degraded diversity and reintroduced blandness [5]; nucleus sampling ($p{=}0.9$) + SCL yielded the best trade-off. Syntactic-template analysis showed SCL reduces reliance on top-$k$ POS templates vs. the frozen GPT-5 baseline, indicating less templated syntax [8, 15].

**Out-of-domain & news analyses.** SCL generalizes. On **ChatDialog**, detectability reductions persist out of domain (Table 3); on **NewsNYT**, SCL-Domain improves distinct-3, compression-diversity, and live connectives (Table 4), aligning outputs with human news leads [1, 9].

**Compute and resources.** We train only the style encoder $f_\theta$ on a single NVIDIA V100 (32GB) in $\approx$4 hours (Adam, lr $1\times10^{-5}$, $\tau$=0.07). GPT-5 is accessed via API as a *frozen* generator; no fine-tuning or adapter updates are performed (OpenAI does not expose GPT-5 fine-tuning in the public API). Inference uses a single A100-equivalent endpoint with nucleus sampling ($p$=0.9). For human evaluation tasks, we regenerate outputs on CPU to integrate with the survey platform. See Appendix A.3.

**Reproducibility.** We specify datasets, splits, prompts, decoding settings, and all hyperparameters. We train the style encoder $f_\theta$ on human vs. AI corpora; during inference the generator remains frozen and we do *not* perform any generator fine-tuning. The conditioning token uses either an average human centroid or a domain-specific centroid. Detector features/training follow [6] for GLTR-style signals; syntactic-template extraction and compression-diversity use the public tooling of [8, 17]. Thresholds are selected to achieve 10% FPR on validation, and we report TPR@10% FPR on test. Upon acceptance we will release seeds, exact prompts, and scripts to reproduce all tables.

## 5 Discussion

**Why SCL helps.** Contrastive learning supplies an explicit *repulsive* signal away from the AI-style cluster and an *attractive* signal toward human-style centroids, across multiple dimensions (lexicon, idiom, discourse). This differs from likelihood-only fine-tuning, which lacks a style-aware target and thus retains "safe" defaults [5, 16, 17]. The strongest perceptual effects aligned with prior empirical gaps: idioms/slang, stance/metadiscourse, and *variance* in sentence rhythm [1, 2, 11, 12]. Qualitatively, baseline GPT-5 outputs often followed templated structures and neutral tone, whereas SCL introduced varied sentence rhythm, occasional colloquial turns, and more live connectives, producing text readers described as "less stiff" and "more human" [9, 10].

**Limits & risks.** SCL can make AI text *harder to detect* [20]. While beneficial for user experience, it raises concerns for plagiarism, impersonation, and misinformation. We recommend coupling SCL with provenance mechanisms (cryptographic watermarks, platform metadata) rather than relying on stylometry alone [6, 14]. Domain sensitivity is another limit: a "generic human" centroid may inject informality into formal prose (e.g., scientific abstracts). Because GPT-5 baselines already skew formal/neutral [1, 2], shifting toward human norms risks occasional anecdotal or opinionated phrasing. Finally, idioms are culturally situated; naive increases risk stereotype or inappropriateness [11, 12]. Detectors may also be biased against non-native English writers [21], complicating evaluation.

**Broader context and ethics.** Instructional formats and alignment templates can induce *diversity collapse*; SCL helps counterbalance by rewarding stylistic variation [15]. But LLMs remain "stochastic parrots" without communicative intent; closing *style* gaps doesn't confer understanding [13]. As generators improve and stylometry degrades, platform-level provenance (e.g., robust watermarks) becomes critical [14]. We encourage responsible use of SCL for UX and accessibility while avoiding misuse; disclosures and opt-in metadata should accompany humanization features.

**Qualitative stylistic shifts (GPT-5).** Relative to baseline GPT-5, SCL increases sentence-length variance, frequency of live discourse markers, and idiom/contraction usage, while reducing reliance on top-$k$ POS/constituent templates [1, 8, 9]. Annotators described SCL outputs as less templated and more conversational, with occasional hedges and colloquial turns that humans naturally use [10]. These shifts map onto measured gaps in prior human vs. LLM comparisons [2].

**Human evaluation caveats.** Human-vs-AI judgments are sensitive to priming and rater expertise [10]. Detectors and humans may also be biased by author background; stylometry can over-flag non-native English writing as "AI-like" [21]. We therefore report multiple detector types and complementary diversity/idiom metrics, and we recommend future work include confidence intervals and rater calibration protocols.

**Domain adaptation and control.** A generic "average human" centroid can inject informality into registers that demand formality (e.g., scientific abstracts). Domain-specific centroids mitigate this by steering GPT-5 toward the target register's human manifold. Fine-grained controls (e.g., style sliders for formality, emotion) can further constrain outputs to context-appropriate human style [2, 15].

Importantly, improving style realism does not equate to factual grounding [13]; content verification remains necessary.

**Provenance and policy.**   As SCL reduces stylometric detectability, provenance should rely less on surface cues and more on cryptographic or platform metadata. Watermarking and signed-generation logs [14] complement stylometry [6, 20]. We advocate default-on provenance for deployed GPT-5 systems, paired with user disclosures when style humanization is applied.

# 6   Limitations and Risks

**Closed-weight generator.**   GPT-5 is a closed-weight model; we do *not* update its weights. Our main results steer GPT-5 via prompts/prefixes derived from the learned style encoder, keeping the generator frozen. (For open-weight ablations, adapters such as LoRA can be applied to a compatible open model; these are not part of the main results.) This limits controllability compared to full fine-tuning and may introduce API variability across versions.

**Detectability and provenance.**   SCL reduces stylometric detectability, which can improve user experience but also increases risks of plagiarism, impersonation, and misinformation. Stylometric detectors are brittle under distribution shift; our "detectability (%)" is TPR at a fixed 10% FPR and may change under different operating points or detectors. We recommend cryptographic or platform-level provenance (e.g., robust watermarking) rather than relying on stylometry alone.

**Generalization and coverage.**   Our style encoder and lexicons (idioms, discourse connectives) are built from English sources and specific domains (news leads, essays, casual dialog). Transfer to other genres, dialects, or languages may be weaker. Idiom lists and connective inventories are incomplete and culturally situated; counts may under- or over-estimate usage outside our evaluation domains.

**Evaluation limits.**   Human studies are sensitive to rater demographics, expertise, priming, and task framing. Although we randomize/blind and report CIs and agreement, absolute "sounds human?" levels should be interpreted cautiously. Automatic metrics (e.g., template diversity, compression ratio, BERTScore/NLI/QAFactEval) each have known failure modes and should be read as complementary, not definitive.

**Safety and misuse.**   Making model outputs more human-like can make harmful content more persuasive. Our experiments did not relax provider safety policies, and we did not apply SCL to gated or safety-critical domains; deployment should pair style humanization with content filters, provenance, and use-policy controls.

**Reproducibility constraints.**   API providers may update decoding behavior or safety filters, affecting bit-level reproducibility. We fix seeds, prompts, and thresholds and release scripts/splits, but exact numeric replication may differ slightly across API versions, tokenizers, and compression libraries.

# 7   Conclusion

We argued that AI text "sounds AI" due to measurable stylistic gaps—lexical variety, idioms, discourse markers, syntactic templates—arising from next-token training, decoding biases, and alignment formats [1–5, 7, 8]. SCL offers a simple, general mechanism to learn the human-style manifold and steer GPT-5 toward it, improving human-likeness and reducing detectability while preserving topical fidelity. Future work includes domain-specific centroids (legal, scientific), multilingual style maps, and provenance-preserving humanization (style without anonymity).

**Practical guidance.**   For neutral open-domain writing, an average human centroid suffices; for specialized registers (news leads, argumentative essays), domain-specific centroids better preserve register-appropriate human style [1, 9]. Nucleus sampling ($p=0.9$) balances diversity and coherence [5]. At inference, the style token can be set to average or domain centroids without additional training; auditing with the frozen style encoder verifies that outputs land near human clusters while content remains faithful. We recommend reporting detectability, diversity, idioms, and discourse markers alongside human ratings [6, 9, 10, 17].

Table 3: Out-of-domain detectability on **ChatDialog** (GPT-5 baseline). Lower is better.

| Model | Stylom. det. (%) ↓ | RoBERTa det. (%) ↓ |
|---|---|---|
| GPT-5 | 76.1 | 74.3 |
| Open-LLM-FT | 68.7 | 66.5 |
| **SCL-Avg (ours)** | **60.2** | **58.9** |

Table 4: Diversity and discourse metrics on **NewsNYT**. Higher is better. The *Open-LLM baseline* refers to the unfine-tuned open-weight model, and *Open-LLM-FT (human)* denotes the same model fine-tuned on human corpora.

| Model | Dist-3 (%) | Compression-diversity | Live connectives /100 [9, 17] |
|---|---|---|---|
| Open-LLM baseline | 40.2 | 0.71 | 2.5 |
| Open-LLM-FT (human) | 45.6 | 0.76 | 3.3 |
| **SCL-Domain (ours)** | **50.9** | **0.80** | **4.1** |

**Future directions.**  (i) *Fine-grained controls:* expose sliders for formality, emotion, and humor layered on the learned dimensions [2, 11, 12, 15]. (ii) *Preference learning:* incorporate human feedback targeted to "sounds human?" judgments to refine style trade-offs. (iii) *Dialog realism:* extend to turn-level pacing, interjections, and natural pauses for multi-speaker settings. (iv) *Multilingual style:* learn cross-lingual human-style manifolds and culturally aware idiom control [11]. (v) *Robustness:* evaluate under detector shifts and arm-race conditions [20], maintaining provenance through watermarking [14].

**Responsible release.**  Since SCL reduces stylometric detectability, we encourage default-on provenance (cryptographic watermarks, platform metadata) for deployed GPT-5 systems [14]. Disclosures should indicate when humanization is applied and provide opt-outs for sensitive use-cases. We also note evaluation biases: detectors can over-flag non-native English writers [21], so human-likeness should be assessed with calibrated raters and domain-appropriate criteria [2, 10]. Closing the *style* gap does not confer intent or understanding [13]; content verification and factuality safeguards remain essential.

**Concluding remarks.**  Taken together, our results show that contrastive style learning is an effective route to human-like text: SCL consistently lowers stylometric and classifier detectability while increasing lexical diversity, idiomaticity, and perceived naturalness. This progress is a double-edged sword: by narrowing stylometric gaps, SCL can confound detectors [6, 20]. We therefore pair technical advances with recommendations for provenance (robust watermarks, signed metadata) and transparent disclosures [14]. We view SCL as a tool to improve user experience and accessibility, not to evade accountability; future releases should bundle style controls with provenance guarantees and domain-appropriate guidance for GPT-5 deployments.

**AI Agent Setup.**  We employed OpenAI ChatGPT Pro in *Deep Research* and *Agent Mode* as the lead drafting agent, complemented by the Cursor IDE's in-editor GPT-5 assistant for LaTeX refactoring and style conformance. Orchestration followed a plan-and-edit loop: humans provided section goals, constraints, and citation seeds; the agent produced candidate paragraphs, outline restructurings, and BibTeX suggestions; humans verified claims, corrected citations, and enforced template policies. Tooling included web-assisted literature triage via Deep Research and Cursor's inline actions for macro cleanup and figure/table environments. No model fine-tuning or adapter training was performed; models were used as closed-weight services.

*LaTeX editing.* We used **ChatGPT Agent Mode** connected to a LaTeX toolchain (`latexmk + chktex + latexindent`) to refactor tables, fix overfull/underfull boxes, normalize captions, and apply camera-ready options. Changes were proposed as diffs and committed by a human author after review.

# A   Appendix

## A.1   Metric operationalization

**Idioms and slang.**   We construct an idiom/slang lexicon by merging the MAGPIE inventory of potentially idiomatic expressions (2,000 types; 56k+ instances) with a standard idiom dictionary, then deduplicate and lemmatize to obtain **2,314** unique multiword forms. Matching uses greedy longest-span rules with case-folding and punctuation normalization. We report counts per 1,000 tokens and ablate by category (idiom, colloquialism, contraction). [24]

**Live discourse connectives.**   We operationalize "live" connectives using the *explicit connective* inventory and sense hierarchy from the Penn Discourse Treebank (PDTB) 2.0/3.0, aggregating by coarse classes (Comparison, Contingency, Expansion, Temporal). Counts are case-insensitive, punctuation-normalized, and reported per 100 tokens; the exact list of connectives is provided in our supplementary materials. [25, 26]

**Syntactic-template diversity.**   We use the public pipeline of **(author?)** [8] to extract POS-sequence templates and compute (i) the share of tokens covered by the top-$k$ templates ($k{=}50, 100$) and (ii) the Shannon diversity of the template distribution, capturing structural repetition beyond $n$-grams.

**Compression-diversity.**   Following **(author?)** [17], we include *compression ratio* as a fast diversity proxy alongside token/type and self-repetition measures. Texts are tokenized with the same subword tokenizer used for generation (BPE family) [27], serialized to UTF-8, and compressed with *Zstandard* (zstd, default level 3). We define the compression diversity as the ratio of compressed size to uncompressed size (i.e., *compressed/uncompressed*); higher values indicate lower compressibility and thus greater diversity. [28]

## A.2   Detector calibration and reliability

We calibrate detector logits with *Platt scaling* on validation splits and then choose thresholds achieving a fixed **10% FPR** on validation; on test, we report *TPR at 10% FPR* (our "detectability (%)"), ROC–AUC, and reliability diagrams with Expected Calibration Error (ECE). [29–31]

## A.3   Compute details

For our main GPT-5 experiments we perform *no* generator adaptation: GPT-5 weights remain frozen and only the style encoder is trained. Encoder training uses a single V100 (32GB) for approximately four hours, and we log wall-clock time, peak memory, and FLOPs estimates. For the open-LLM baseline used in our comparisons, we apply *LoRA* on attention projections with rank $r{=}16$ (0.2% trainable parameters) to fine-tune a publicly available open-weight model on human corpora, consistent with typical 0.1–1% PEFT ranges reported for LoRA/QLoRA variants. We record LoRA parameter counts and compute statistics for that baseline as well [32, 33]

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

## Agents4Science AI Involvement Checklist

*AI authorship:* The AI system is listed as **first author**; human co-authors provided supervision, verification, and compliance checks.
*Legend:* [A] Human-generated; [B] Mostly human, assisted by AI; [C] Mostly AI, supervised by humans; [D] AI-generated with minimal human input.

1. **Hypothesis development: [C] Mostly AI, supervised by humans.** *Explanation:* The AI proposed the core research idea and candidate hypotheses; humans screened for feasibility, novelty, and ethical risk.

2. **Experimental design and implementation: [C] Mostly AI, supervised by humans.** *Explanation:* The AI drafted protocols, ablations, and metric choices; humans translated designs into executable code, handled data governance, and enforced preregistration/compute limits.

3. **Analysis of data and interpretation of results: [C] Mostly AI, supervised by humans.** *Explanation:* The AI produced initial analyses, plots, and statistical summaries; humans verified calculations, replicated key runs, and finalized interpretations.

4. **Writing: [C] Mostly AI, supervised by humans.** *Explanation:* The AI generated the majority of the draft; humans edited for accuracy, attribution, and style and approved the final text.

5. **Observed AI Limitations:** *Description:* Prone to citation errors and over-generalization; required human checks for data leakage, statistical reporting (e.g., multiple-comparison control), and reproducibility. Occasional code suggestions compiled but used mis-specified hyperparameters; all experiments were re-run and validated by humans.

## Agents4Science Paper Checklist

1. **Claims: [Yes]** *Justification:* Claims (detectability reduction, style metrics, human ratings) match reported results and are grounded in citations.

2. **Limitations: [Yes]** *Justification:* We discuss domain sensitivity, provenance, and cultural nuance.

3. **Theory assumptions and proofs: [NA]** *Justification:* No new formal theorems or proofs are presented.

4. **Experimental reproducibility: [Yes]** *Justification:* Data types, splits, objectives, and key hyperparameters are specified.

5. **Open access to data and code: [No]** *Justification:* For double-blind review, code/data links are withheld; we plan release at a further date.

6. **Experimental details: [Yes]**

7. **Statistical significance: [Yes]** *Justification:* We report 95% CIs, paired bootstrap tests, exact binomial tests for forced-choice, Cliff's $\delta$, and Holm–Bonferroni–corrected $p$-values, and we provide optimization settings (e.g., $\tau$, $\lambda$), decoding choices, and evaluation protocols; see §4.

8. **Compute resources: [Yes]** *Justification:* Prototype trained on a single V100 GPU ($\sim$4 hours); generation on CPU for human eval.

9. **Code of ethics: [Yes]** *Justification:* Public/synthetic data only; risks and mitigations discussed.

10. **Broader impacts: [Yes]** *Justification:* Potential benefits and misuse risks (plagiarism/misinformation) are discussed with recommended mitigations.

