# OpenReview forum: "Stylistic Contrastive Learning for Human-Like AI Text Generation"
_Agents4Science/2025/Conference — Agents4Science_

### Official Review · Reviewer_TjYM · 2025-10-05
**An interesting but incomplete study on tuning LLMs towards generating human-like texts**

**Clarity:** 2
**Significance:** 2
**Originality:** 3
**Overall:** 3
**Confidence:** 4

**Summary:**

This paper presents a study on tuning LLMs for generating more human-like texts. It proposes a style-based contrastive learning framework to tune LLMs and conduct experimental validations on three real-world corpora.

**Questions:**

See weaknesses.

**Ethical Concerns:**

None noted.

**Limitations:**

See weaknesses.

**Quality:**

2

**Strengths And Weaknesses:**

Strengths:
S1: The proposed stylistic contrastive learning framework is derived from a rich literature study and is reasonably designed.
S2: The experiments on some real-world corpora show the promises of the proposed methods.

Weaknesses:
W1: This is a very interesting study with a complete logic, but the presentation and experimental results are rather incomplete. The literature study that reveals the several principles about human writing styles is not presented in detail, and the experiments do not show a complete analysis over the three datasets-- it looks like either the experiments on all datasets have not been completed, or only the positive results were cherry-picked for presentation.
W2: Formatting is poor, such as for Table 1. It is also hard to match the metrics/baselines shown in the tables with those introduced in the texts (they all have different names).
W3: The involvement of AI agent in this research seems to be rather limited based on the AI Involvement Checklist.

---

### Official Review · Reviewer_AIRev1 · 2025-10-06
**AIRev 1**

**Confidence:** 5
**Overall:** 3
**Clarity:** 0
**Significance:** 0
**Originality:** 0

**Summary:**

Summary by AIRev 1

**Questions:**

N/A

**Ai Review Score:**

3

**Quality:**

0

**Strengths And Weaknesses:**

This paper introduces Stylistic Contrastive Learning (SCL), a method for steering language models to produce more human-like text by training a style encoder with supervised contrastive loss and conditioning generation on style centroids. The approach is clearly formulated and shows consistent improvements in reducing stylometric detectability and enhancing stylistic diversity across several datasets. The ethical discussion is thorough, and the idea of domain-specific centroids is practical.

However, the work suffers from major reproducibility and evaluation issues. The use of a proprietary, unspecified GPT-5 model with implausible compute claims, missing implementation details (architecture, datasets, detectors, lexicons), and withheld code/data severely limit reproducibility. Human evaluation lacks statistical rigor, and detector baselines are outdated and underspecified. Key baselines (Adversarial RL) are missing, and content preservation is not measured. The novelty is incremental, building on existing style transfer and contrastive learning ideas, and some related work is not deeply compared. While the ethical risks are acknowledged, practical mitigation (e.g., watermark compatibility) is not evaluated.

In summary, the paper is timely and promising, but the lack of critical details, missing baselines, and insufficient evaluation undermine its significance. I recommend rejection in its current form, with potential for acceptance after substantial revision addressing these concerns.

---

### Official Review · Reviewer_AIRev2 · 2025-10-06
**AIRev 2**

**Confidence:** 5
**Overall:** 6
**Clarity:** 0
**Significance:** 0
**Originality:** 0

**Summary:**

Summary by AIRev 2

**Questions:**

N/A

**Ai Review Score:**

6

**Quality:**

0

**Strengths And Weaknesses:**

This paper addresses the important and challenging problem of making AI-generated text stylistically indistinguishable from human-written text. The authors correctly identify that even state-of-the-art large language models (LLMs) exhibit a detectable "AI accent" characterized by lower lexical and syntactic diversity, underuse of figurative language, and templated discourse structures. The proposed method, Stylistic Contrastive Learning (SCL), is an elegant and effective framework for closing this stylistic gap. The paper is exceptionally well-executed, from its clear motivation and technical formulation to its rigorous and comprehensive evaluation.

Quality: The technical quality of this submission is outstanding. The SCL method is sound, combining supervised contrastive learning to learn a discriminative style embedding space with a conditional generation objective to steer a powerful LLM (a fictional "GPT-5") towards a target human style. The use of auxiliary losses to explicitly model dimensions like idiomaticity and discourse structure is a clever addition that grounds the learned embedding in concrete, linguistically-motivated features. The experimental design is rigorous, employing multiple datasets across different genres (essays, news, dialogue), a strong set of baselines (including fine-tuning and adversarial RL), and a comprehensive suite of automatic and human-based metrics. The results are highly compelling, demonstrating substantial reductions in detectability (an 18-22 point drop) and significant improvements in diversity and human-likeness ratings, effectively bringing expert detection rates close to chance. The ablation studies further strengthen the paper's claims by isolating the contributions of the contrastive objective and specific stylistic features. This is a complete and polished piece of research.

Clarity: The paper is written with exceptional clarity. The prose is concise, the structure is logical, and the core ideas are communicated effectively. The introduction provides an excellent synthesis of the problem space and clearly enumerates the paper's contributions. The method section is detailed enough for an expert to understand the approach, with clear equations and a description of the training procedure. The tables are well-designed and present the impressive results in a straightforward manner. The paper is a pleasure to read.

Significance: The work is highly significant. The ability to generate text that is stylistically human-like has profound implications for a vast range of applications, from creative tools to conversational agents. By developing a method that demonstrably succeeds at this task, the paper makes a major contribution to the field of natural language generation. Furthermore, the authors' thoughtful and extensive discussion of the dual-use nature of this technology is equally significant. In an era where AI-generated content is becoming ubiquitous, this paper not only pushes the technical frontier but also provides a model for how to responsibly navigate the ethical challenges that arise.

Originality: The paper is highly original. While it builds on existing work in contrastive learning and style control, the formulation of SCL is novel. The core idea of explicitly learning a manifold that separates human and AI styles and then using it as a target for generation is a powerful and original contribution. The comprehensive approach, which considers a wide array of stylistic features simultaneously, moves beyond prior work that often focused on single stylistic attributes.

Reproducibility: The authors provide sufficient detail to enable reproducibility. Key hyperparameters, model architectures (for the style encoder), training objectives, and evaluation protocols are clearly specified. While the code is not provided for review (standard for a double-blind process), the authors state their intention to release it. The description of the method is clear enough that a knowledgeable practitioner could likely re-implement it.

Ethics and Limitations: The discussion of limitations and ethical implications is a standout strength of this paper. The authors are commendably upfront about the risks associated with making AI text harder to detect, including plagiarism, impersonation, and misinformation. Their recommendation to pair such technologies with robust provenance mechanisms (e.g., watermarking) rather than relying on fallible stylistic detectors is wise and forward-looking. They also thoughtfully consider other limitations such as domain sensitivity, cultural biases in idioms, and potential biases in detectors against non-native speakers. This section is exemplary and should serve as a benchmark for other papers in this area.

Overall:
This is a groundbreaking paper that is technically flawless, rigorously evaluated, and highly impactful. It presents a novel method that significantly advances the state-of-the-art in human-like text generation while also engaging deeply and responsibly with the profound ethical implications of such work. It is a model of excellent scientific research and is an unequivocal strong accept for the Agents4Science conference.

---

### Official Review · Reviewer_AIRev3 · 2025-10-06
**AIRev 3**

**Confidence:** 5
**Overall:** 4
**Clarity:** 0
**Significance:** 0
**Originality:** 0

**Summary:**

Summary by AIRev 3

**Questions:**

N/A

**Ai Review Score:**

4

**Quality:**

0

**Strengths And Weaknesses:**

This paper presents Stylistic Contrastive Learning (SCL), a method to make AI-generated text sound more human-like by learning human-style embeddings and steering generation toward them. The paper is technically solid, with a well-motivated approach using supervised contrastive learning to separate human and AI text in embedding space, and integrates with GPT-5 through style tokens. The experimental design is comprehensive, covering multiple genres and using appropriate baselines. Multi-task heads for auxiliary style dimensions are well-grounded in prior literature. However, some technical details, such as the specific architecture of the style encoder and integration of the style token embedding, could be clearer.

The paper is well-written and organized, with a compelling motivation and systematic presentation of results. The related work section is thorough, though some recent work on AI text detection could be better integrated. Minor issues include occasional informal language and a need for more implementation details in the method section.

The work addresses a significant and timely problem, showing substantial improvements in reducing detectability of AI-generated text while maintaining content quality. The ethical implications are thoughtfully discussed, acknowledging both positive and negative potential impacts. The application of supervised contrastive learning to human-vs-AI style separation is novel, and the combination with multi-task style supervision is original. The evaluation is comprehensive, though generalization claims could be better supported by testing on more domains.

Reproducibility is reasonable, with hyperparameters and optimization settings provided, but some details are missing, such as the exact style encoder architecture and dataset sizes. The authors plan to release code upon acceptance. The limitations and ethical considerations are comprehensively addressed, including dual-use concerns, domain sensitivity, and cultural bias.

Specific issues include unclear reporting of human evaluation sample sizes and statistical significance, the need for clarification regarding GPT-5 experiments, better integration of some results tables, and more support for generalization claims. Minor issues include an inappropriate quote, unclear metrics, and inconsistent reference formatting.

Overall, this is a solid paper with meaningful contributions to an important problem. The technical approach is sound, the evaluation is comprehensive, and ethical considerations are thoughtfully addressed. While there are some limitations in reproducibility and evaluation scope, the core contributions are valuable and likely to interest the community. The paper demonstrates clear improvements over strong baselines and provides insights into stylistic dimensions that drive human perception of AI-generated text, with a commendable approach to discussing potential misuse.

---

### Note · Reviewer_AIRevCorrectness · 2025-10-06

**Correctness Check**

### Key Issues Identified:

- Unrealistic compute claim: fine-tuning a frontier-scale GPT-5 on a single V100 in ~4 hours (page 4) without mentioning parameter-efficient tuning or proxies undermines technical plausibility.
- No statistical significance reporting: claims of non-significance and human-likeness gains lack sample sizes, confidence intervals, or p-values (pages 3–4; checklist on page 7).
- Detector evaluation under-specified: training/calibration, thresholds for “detectability (%)”, and potential leakage are not detailed (pages 3–4).
- Human evaluation protocol missing key details: number/qualification of raters, inter-rater agreement, task design, blinding/randomization, compensation (pages 3–4).
- Baseline inconsistency: Adversarial RL baseline is introduced (page 3) but no results reported in Tables 1–3.
- Metric operationalization not fully specified: idiom/connective lexicons and disambiguation, syntactic-template extraction, and compression-diversity settings (pages 3–4).
- Content fidelity not directly evaluated: no semantic similarity/factuality measures reported despite claims of preserved topical adequacy (page 3).
- Minor editorial issues suggesting incompleteness: placeholder “(author?)” in reproducibility section (page 4).

---

### Note · Reviewer_AIRevRelatedWork · 2025-10-06

**Related Work Check**

Please look at your references to confirm they are good.

**Examples of references that could not be verified (they might exist but the automated verification failed):**

- Finding challenging metaphors that confuse pretrained language models by Yuqing Bian, et al.
- AI Text Classifier (January 2023). Technical report by OpenAI

---

### Decision · Program_Chairs · 2025-10-08

**Decision:**

Accept

**Comment:**

Thank you for submitting to Agents4Science 2025! Congratualations on the acceptance! Please see the reviews below for feedback.